# Influence of Parameters Used to Prepare Sterile Solutions of Poloxamer 188 on Their Physicochemical Properties

**DOI:** 10.3390/polym17010062

**Published:** 2024-12-29

**Authors:** Alexander Yegorov, Sergei Pushkin, Elena Arshintseva, Maxim Molchanov, Maria Timchenko

**Affiliations:** 1Institute of Theoretical and Experimental Biophysics RAS, Pushchino 142290, Russia; a.yeg0rov@yandex.ru (A.Y.); lvlaks.m@gmail.com (M.M.); 2Medical Emulsions LLC, Serpukhov, Pushchino 142290, Russia; laboratorypushkin@yandex.ru (S.P.); micolina@mail.ru (E.A.)

**Keywords:** poloxamer 188, atomic force microscopy, viscometry, NMR spectroscopy, physicochemical properties

## Abstract

The physicochemical properties of emulsions based on poloxamers (triblock copolymers of a hydrophobic polyoxypropylene chain and two hydrophilic polyoxyethylene chains) depend on the composition and preparation method. This study examined the impact of poloxamer P188 concentration, autoclaving mode, heating, and salt presence on the viscosity, particle size distribution, and morphology of particles using viscometric analysis, dynamic light scattering (DLS), and atomic force microscopy (AFM). It was shown that sample preparation affects the particle size and morphology but not the chemical composition of P188. The most similar properties were found for 10% P188 samples sterilized by filtration and autoclaving. The higher autoclave temperature and additional heating of the 10% P188 samples to 70 °C resulted in the formation of larger particles. For 4% P188 samples with 0.6% NaCl, samples heated at 70 °C for 15 h after sterilization filtration and autoclaving were the most similar and homogeneous. The 4% P188 sample with the higher autoclave temperature and subsequent heating had the lowest viscosity. In contrast to 10% P188, for 4% P188 in the presence of salt, the lack of heating resulted in the formation of large particles. The 4% P188 solutions with NaCl were more stable during storage than those with a higher concentration.

## 1. Introduction

Since the 1970s, studies on the development of artificial blood substitutes based on perfluoroorganic (PFO) compounds in the USA, Japan, and the USSR have resulted in more than 20 new PFO emulsions, which have been presented to the world scientific community as both blood substitutes and drugs for PFO therapy of other nosologies. Various PFO compounds or their mixtures were used in the composition of the emulsions created: Perfluorodecalin (PFD), Perfluorotributylamine (PFTBA), Perfluorotripropylamine (PFTPA), Fozhalin, Perfluoromethylcyclohexylpiperidine (PMCP), Perfluorooctylbromide (PFOB), Perfluorodecylbromide (PFDB), Perfluorodichlorooctane, and Perfluoro-tert-butylcyclohexane (FtBu). The emulsifiers used were poloxamer 188 (P188), phospholipids derived from egg yolk, glycerol and triglycerides, and various fluorinated surfactants [1,2]. However, only two blood substitutes based on PFO emulsions have received the status of a registered drug in World Health Organization (WHO) systems: the Japanese–American drug Fluosol-DA and Perftoran in Russia since 1996 [2]. And, it is quite possible that this was made possible by the fact that P188 was used as an emulsifier for these drugs. It should be clarified that the emulsifier for Fluosol-DA was a mixture of P188, egg yolk phospholipids, and glycerol [1,2]. In 2005, the artificial blood substitute Perftoran was approved for clinical use in Ukraine, Kazakhstan, Kyrgyzstan, and Uzbekistan, and in Mexico as Perftec [1,2].

Poloxamer 188 (Pluronic F68 or Kolliphor P188) is a triblock copolymer with an average molecular weight of 8000 Da. Triblock copolymers composed of poly(ethylene oxide)/poly(propylene oxide)/poly(ethylene oxide) with surfactant properties are widely used in the pharmaceutical industry for stabilization and emulsification and as adjuvants. Due to their lower toxicity and critical micelle concentration (CMC), they are preferred over ionic surfactants [3]. Triblock’s unique composition of two hydrophilic poly(ethylene oxide) groups and a central hydrophobic poly(propylene oxide) group gives it amphiphilic properties. The self-assembly characteristics of these copolymers, such as micelle size, CMC, and micelle temperature, can be adjusted by varying the ratio of ethylene oxide (EO) to propylene oxide (PO) and the chain lengths [4].

Pharmaceutical formulations containing triblock copolymers in aqueous solutions with buffers and salts can affect the association of the copolymers. The presence of inorganic salts influences the structure of micelles and their solubilization, impacting both the PO and EO groups through competition for solvation. Salts decrease the critical temperature for micelle formation by reducing the hydration of PO groups, while also affecting hydrogen bonds between PEO and water and altering PEO solubility [5,6]. Therefore, it is important to consider the effects of additives and the drug manufacturing process on the physicochemical properties of triblock copolymers.

Apart from being used as an adjuvant for the production of many pharmaceuticals and as an emulsifier for PFO emulsions (poloxamers are included in the American and European *Pharmacopoeias*), P188 is now increasingly being used as an API. It has potential applications in stabilizing and repairing cell membranes damaged by injury or disease. P188 has been shown to protect neuronal and non-neuronal cells, improve cell function after exposure to various stressors, and enhance blood characteristics with minimal side effects [7,8,9]. Comparative studies by the authors on the chronic and acute toxicity of intravenous administration of sterile aqueous solutions of P188 from different manufacturers showed their low toxicity and similarity [10,11]. This contradicts previous conclusions on the toxicity of P188 [1].

It is known that the use of PFO emulsions has a positive effect on the function of the liver [12,13,14,15]. To date, the authors have published the results of preclinical studies on the use of the new PFO emulsion Oxyphtem (composition: PMCP and P188) under conditions of experimental systemic inflammatory response for reparative regeneration and adaptation of the liver [16], the results of its toxicological comparison with Perftoran emulsion [17], and the results of a study on the efficacy of PFO emulsions in stimulating hair growth in C57BL/6 mice [18].

The main stage of the technological process for the production of PFO emulsions with particle sizes up to 150 nm (Perftoran, Oxyphtem) is homogenization under pressures up to 60.8 MPa, during which emulsification of sterile PFO compounds takes place in a sterile aqueous solution of P188 under aseptic conditions, as the final product is not sterilized [19]. The final step in the preparation of a PFO emulsion, e.g., Perftoran, is to adjust the concentration of PFO compounds in the emulsion to 10% by volume and poloxamer to 4% using a water–salt composition [17,19]. Different authors suggest the following steps to prepare an aqueous solution of P188: autoclaving at 121 °C for 12–15 min and heating at 65–75 °C for 12–16 h [20], heating at 70 °C for 8–24 h [21], or autoclaving at 118 °C for 12–15 min only [22].

Assuming that the reactogenicity and toxicity of PFO emulsion are influenced not only by the composition of the emulsion, but also by the manufacturing process, including the sterilization of its constituents, this study investigates the changes in the physicochemical properties of P188 solutions under different preparation conditions, including sterilization procedures, heating time, concentration, and the presence of salts. The viscosity, homogeneity, particle size, morphology, and chemical composition of the solutions were analyzed using viscometry, dynamic light scattering, atomic force microscopy, and ^1^H NMR spectroscopy, respectively.

## 2. Materials and Methods

### 2.1. Sample Preparation

Kolliphor P188 (GND10221B series, BASF, Germany) was used as poloxamer 188.

A laboratory stirrer, type PW (500 ÷ 700 rpm), a VK-75 autoclave, and an ShS-80 dry heat oven were used to prepare the samples.

To prepare a 10% solution of P188 (samples 1–7), 110 g of P188 was gradually added to 0.5 L of water for injection and stirred until completely dissolved. The volume of the solution was adjusted to 1.1 L with water for injection. The P188 solution was filtered under aseptic conditions at a pressure of 68.65–147.1 kPa (0.7–1.5 at.) through a sterile hydrophilic membrane filter with a pore diameter of 0.20–0.22 μm (Millipore, Burlington, MA, USA). The filtrate was collected in sterile smooth-necked bottles for infusion preparations with a capacity of 100 mL according to GOST 10782-85, 50 mL each, which were then sealed with sterile rubber stoppers, covered with sterile aluminum caps, and rolled on a form–fill–seal machine, ZPR-00-0. The total yield was 20 bottles of 50 mL P188 solution. Two bottles were allocated to each group and were numbered from 1 to 7.

Bottles numbered 3, 4, and 7 were sterilized at a temperature of 118 °C and a pressure of 88.26 kPa (0.9 at.) for 12 min. After autoclaving, the bottles with solution were cooled at room temperature (18–25 °C). Bottles numbered 5 and 6 were sterilized at 121 °C and 107.9 kPa (1.1 at.) for 12 min. After autoclaving, the bottles with solution were cooled at room temperature (18–25 °C). Afterward, bottles numbered 2, 4, and 6 were heated in a dry heat oven at 70 °C for 15 h, and bottles numbered 7 for 2 h.

To prepare a 4% solution of P188 with 0.6% NaCl (samples 8–13), 44 g of P188 was gradually added to 0.5 L of water for injection and stirred until completely dissolved, then 6.6 g of NaCl was added and stirred. The volume of the solution was adjusted to 1.1 L with water for injection. Filtration and bottling were carried out as previously described for 10% P188. The total output was 20 bottles of 50 mL P188 solution with the salt composition. Each group received 2 bottles. The bottles were numbered from 8 to 13.

Bottles numbered 10 and 11 were sterilized in a VK-75 autoclave at a temperature of 118 °C and a pressure of 88.26 kPa (0.9 at.) for 12 min. After autoclaving, the bottles with solution were cooled at room temperature (18–25 °C).

The bottles numbered 12 and 13 were sterilized in an autoclave at 121 °C and 107.9 kPa (1.1 at.) for 12 min. After autoclaving, the bottles with solution were cooled at room temperature (18–25 °C).

Afterward, bottles numbered 9, 11, and 13 were heated in a dry heat oven at 70 °C for 15 h.

The sample for analysis was taken using a syringe by puncturing the rubber cap. Samples were stored at room temperature for 14 months.

### 2.2. Viscometric Analysis

Dynamic viscosity measurements were performed using an SV-10A A&D vibroviscometer (Japan) at 20 °C in 10 mL cuvettes. The vibroviscometer was calibrated with distilled water at 20 °C. When using the vibration method to measure viscosity, the density of the sample significantly affects the result. The device’s display shows a value equal to the product of the dynamic viscosity and the sample density (mPa⋅s*g/cm^3^). Therefore, to obtain true values of dynamic viscosity, the density of the samples was determined using the pycnometric method. Mass measurements were carried out using an analytical balance (Sartorius CP64). The density was calculated according to the following formula:$$\rho_{solution}=\frac{m_{2}-m}{m_{1}-m}\rho_{water}$$
where $\rho_{solution}$ is the density of the solution, g/cm^3^;

m—pycnometer weight, g;

$m_{1}$—weight of the pycnometer with distilled water, g;

$m_{2}$—weight of device with test solution, g;

$\rho_{water}$—density of distilled water, g/cm^3^.

### 2.3. DLS Measurements

Dynamic light scattering (DLS) was used to determine the particle size distribution of the samples. Measurements were performed on a Malvern Zetasizer NanoZS instrument with a laser power of 4 mW at a backscatter angle of 173° at 37 °C. The sample (100 μL) was placed in a cuvette and then in the instrument. Multiple scans of the sample were taken every 100 μs for 60 s. Cumulative analysis with correlation functions was used to calculate the average diffusion coefficient, from which the particle size (hydrodynamic radius) was estimated using the Stokes–Einstein formula. All calculations are performed automatically by the Malvern system, which determines the size by first measuring the Brownian motion of the particles in the sample using DLS and then interpreting the size obtained using established theory. Mathematical processing and visualization of the data was performed using Zetasizer 7.11 software (Malvern, UK).

### 2.4. AFM Measurements

To prepare samples for AFM, 2 µL of solution was taken, and the solution was transferred to freshly cleaved mica and incubated for 5 min. The sample was then washed three times in a drop of distilled water for 30 s and dried in the air. All studied samples were prepared similarly, which were subsequently sampled depending on the incubation time and loaded on mica.

AFM imaging was performed with an AFM Ntegra-Vita microscope (“NT-MDT”, Moscow, Russia) in noncontact (tapping) mode in air. The typical scan rate was 1 Hz. Measurements were carried out using NSG03cantilevers with a resonance frequency of 47–150 kHz and an ensured 10 nm tip curvature radius. The processing and presentations of AFM images were performed using Nova 1.0.26 software (“NT-MDT”, Moscow, Russia) and Gwyddion 2.50 software (http://gwyddion.net/ accessed on 17 April 2018, Czech Republic). Statistical analysis was carried out in Gwyddion 2.44. Statistical values included the main characteristics of the distribution of height values, including their minimum, maximum, and median values. These data were then used for multivariate statistical analysis using the principal components method.

### 2.5. Multivariate Statistical Analysis

Experimental data were processed and visualized using the R language (version 4.4.0 (2024-04-24) “Puppy Cup”), with the Tidyverse package (version 2.0.0) in the RStudio environment (version 2024.09.0+375 “Cranberry Hibiscus”). Graphs were plotted using ggplot2 (version 3.5.1). Confidence ellipses were plotted using the stat_ellipse() function with a level of 0.95.

### 2.6. ^1^H NMR Analysis

One-dimensional (1D) 1H-NMR spectra were acquired with a Bruker Avance III 600 spectrometer (The Core Facilities Centre of Institute of Theoretical and Experimental Biophysics of the RAS) operating at a frequency of 598.95 MHz (^1^H) and a probe temperature of 298 K. To suppress the signal from water protons, a pre-saturation method was used by applying a 1D pulse sequence, ZGPR. The number of accumulations was 64 scans. The chemical shifts were assigned according to the TSP signal at 0.00 ppm, which acts as an internal reference.

## 3. Results

### 3.1. Preparation of Solutions with Different Concentrations of Poloxamer 188 and Sodium Chloride and Various Sterilization Process Parameters

To study the effects of poloxamer 188 concentration, the presence of salt, and the sterilization mode on the physicochemical properties of the solution, the samples were divided into groups: 1–7—samples containing 10% (*w*/*w*) P188; and 8–13—samples containing 4% (*w*/*w*) P188 and 0.6% (*w*/*w*) NaCl. For samples 1 and 8, only sterilizing filtration was used; for samples 2 and 9, sterilizing filtration was followed by heating for 15 h at 70 °C. The preparation of samples 3 and 10 included the step of sterilizing filtration followed by autoclaving at 0.9 at. (118 °C) and preparations 5 and 12, sterilizing filtration followed by autoclaving at 1.1 at. (121 °C). Samples 4 and 11 were prepared by sterilizing filtration followed by autoclaving at 0.9 at. (118 °C) and further heating for 15 h at 70 °C. Samples 6 and 13 were obtained by sterilizing filtration followed by autoclaving at 1.1 at. (121 °C) and further heating for 15 h at 70 °C. Sample 7 was obtained by sterilizing filtration followed by autoclaving at 0.9 at. (118 °C) and further heating for 2 h at 70 °C. The samples that were analyzed are shown in Table 1.

### 3.2. Analyzing the Viscosity of P188 Samples

Sample groups 1 to 7 and 8 to 13 varied in P188 content. Groups 1 to 7 were more concentrated, containing 10% P188, whereas groups 8 to 13 contained 4% P188. It is obvious that the viscosity of the second group should be lower. To investigate how viscosity changes during sample storage, viscosity was measured on samples after 14 months. The data on the viscosity are given in Table 2.

Long-term storage resulted in a significant decrease in viscosity (by 15–21% of the initial value) in samples 3–6 (10% P188) and 12 (4%P188 + 0.6%NaCl), which were autoclaved. In the remaining cases, viscosity decreased slightly (by 2–10% of the initial value) and increased slightly in sample 2 (by 2% of the initial value).

### 3.3. Dynamic Light Scattering (DLS) Particle Size Analysis of the Samples

Dynamic light scattering (DLS) was used to analyze the particle size distribution of the samples. The studies were carried out one week after sample preparation and after 14 months of storage to investigate changes in particle composition. The data are shown in Table 3. Particle size distribution plots for the one-week samples are represented in the Appendix A for samples 1–8 and Appendix A for 9–13). Similar data were obtained for the samples after 14 months of storage (Appendix A for samples 1–8 and Appendix A for 9–13).

The main part of the particles in all samples are particles with a size of about 4 nm, but there are also particles with a size of about 300 nm (up to 20%), as can be seen from the results obtained. The formation of high-molecular-weight aggregates (up to 4%) was observed in samples 1, 2, 6, 7, and 12.

Long-term storage resulted in changes in particle size distribution. For concentrated samples 2 to 5, there was a 1.5 to 2-fold decrease in the proportion of small particles, with a concomitant increase in the formation of larger aggregates. For 4% P188 solutions in the presence of salt, a slight decrease in the proportion of small particles and an increase in the proportion of aggregates was also observed, particularly for sample 11. Samples 1 and 8 remained almost unchanged, with about 3% more of the smaller aggregates appearing in sample 1 than initially present, whereas in sample 8, 3% more of the larger particles appeared. Sample 6 showed a slight increase (10%) in the proportion of small particles and a decrease in the proportion of large aggregates, while sample 7 showed a slight decrease in the proportion of small particles and an increase in the proportion of aggregates.

### 3.4. Analysis of Particle Morphology in Samples Using Atomic Force Microscopy (AFM)

The morphology and size of the particles present in the samples were studied using the AFM method in semi-contact mode. The studies were carried out one week after sample preparation and after 14 months of its storage to investigate changes in the morphology of particles. The results obtained are shown in Figure 1 and Figure 2, respectively.

As can be seen from the results obtained, which correlate with previously known data for poloxamer F68 [7], the particles present in the samples mostly form dendritic structures of about 4 nm in height on the mica. Large aggregates of about 300–500 nm are found in samples 1, 7, and 12, and aggregates of up to 100 nm in samples 6 and 8. Sample 9 is different from the other samples in that it forms a network of particles up to 200 nm in height on the mica.

The investigation of particle morphology in samples after long-term storage showed that concentrated samples, except samples 1 and 7, contained small aggregates. Samples 5 and 6 contained more sizeable aggregates. For 4% poloxamer solutions in the presence of salt, small aggregates were observed only in sample 11. The main components of all samples were dendritic structures up to 4 nm in height.

### 3.5. Analysis of Differences Between Samples Using Multivariate Statistics

To assess whether there was a significant difference between the one-week samples, all the results obtained using the different methods were entered into a common table and analyzed using multivariate statistical analysis using Principal Component Analysis (PCA). PCA is the most popular unsupervised statistical method for processing data and requires no prior assumptions or knowledge. Unsupervised methods are usually the first step in data analysis, helping to visualize data and identify similarities and differences between study groups. PCA allows you to reduce the dimensionality of the data, remove noise, and identify patterns. Figure 3 and Figure 4 show PCA data.

As expected, samples 1–7 and 8–13 form separate groups. Group 1–7 is more homogeneous than group 8–13. Within each group, there are separate sub-groups indicating the similarity of samples: sub-group 3, 5, 7; sub-group 4, 6; sub-group 10, 12, and sub-group 11, 13. All other samples vary considerably.

As can be seen from the loading plot (Figure 4), the main contribution to the PC2 principal component is made by the temperature of autoclaving, and to PC1 by viscosity and density, which are correlated with each other, as well as DLS data on particle size. At the same time, heating has almost no effect on the principal components of PC1 and PC2, but has a strong effect on the principal component of PC3.

### 3.6. 1H NMR Spectroscopy to Study the Influence of Preparation Parameters on the Chemical Composition of Poloxamer 188

To exclude changes in structure and composition that may occur during sterilizing filtration, autoclaving at different temperatures, and subsequent heating, we obtained ^1^H NMR spectra for samples prepared by sterilizing filtration only (sample 1), autoclaving at 1.1 at. (sample 2), autoclaving at 0.9 at. (sample 3), and autoclaving at 1.1 at. followed by heating at 70 °C for 15 h (sample 4) (Figure 5). The concentrations of the samples for NMR studies were 4% (*w*/*w*).

As can be seen from the spectra, sterilizing filtration, autoclaving at different temperatures, and subsequent heating do not significantly affect the composition of the poloxamer. However, it should be noted that preparations 1 and 3 appear to contain aggregates, which cause a strong broadening of the signal in the spectra at 3.73 ppm. Preparation 3 contains more of these aggregates. The content of aggregates in preparation 2 is much lower, as can be seen from the smooth and symmetrical signal. The most homogeneous preparation is preparation 4, obtained by autoclaving at 1.1 at. followed by heating at 70 °C for 15 h, in which a symmetrical signal is observed. These results correlate with data obtained by other methods.

## 4. Discussion

The stability of the resulting PFO emulsions is the main problem in the manufacture of blood substitutes based on perfluoroorganic compounds. Emulsifiers, including poloxamer 188, are used to solve this problem. It has been found that the emulsion is more stable if the technological process for preparing a solution of poloxamer 188, apart from the stages of sterile filtration and autoclaving, also involves a heating stage at 70 °C for 15 h. Salt is known to affect the properties of micelles. Another poloxamer, Pluronic P85, forms two types of micelles—small and large. A small micelle has a diameter of about 5 nm and is considered a monomolecular micelle. A large micelle has a diameter of about 20 nm and is a polymolecular micelle. The addition of NaCl increases the volume ratio of polymolecular micelles and the effect of inorganic salts on the formation of polymolecular micelles in the following decreasing order: Na_2_HPO_4_ > NaH_2_PO_4_ > NaCl >NaBr, which corresponds to the Hofmeister series [3].

In order to take into account the effect of the presence of salt on the physicochemical properties of poloxamer 188 during its preparation, some samples contained 0.6% NaCl (*w*/*w*). In this work, the effect of P188 concentration, two autoclaving modes (0.9 and 1.1 at.), and heating time after autoclaving on the properties of P188 was analyzed. Multivariate statistical analysis showed that samples 3, 5, and 7 from the group where the P188 concentration was 10% had the most similar properties. For samples 3 and 5, the technological preparation process included sterilization filtration and autoclaving at 0.9 and 1.1 at., respectively, but without further heating. The difference between sample 7 and sample 3 was that sample 7 was subsequently heated at 70 °C for 2 h. The particle morphology and size were the same for preparations 3, 5, and 7, but 7 contained a small amount of large particles (up to 4%), most likely caused by heating. Preparations 4 and 6 also had similar properties; in addition to autoclaving at 0.9 and 1.1 at., respectively, they were heated at 70 °C for 15 h. Preparations 4 and 6 differed little in particle size and morphology, but preparation 6 contained high-molecular-weight aggregates (up to 17%), which may be due to a higher temperature during autoclaving. Samples 11 and 13 from the group containing 4% P188 and 0.6% NaCl also showed very similar characteristics. Their preparation included the steps of sterilization filtration and autoclaving at 0.9 and 1.1 at., respectively, followed by heating at 70 °C for 15 h. The particle size distribution and morphology were almost identical. A small number of small aggregates in the AFM image of sample 13 may be due to the presence of salt on the mica. At the same time, preparation 13 had the lowest viscosity. Preparations 10 and 12 also correlated with each other; their preparation only included sterilization filtration and autoclaving at 0.9 and 1.1 at., respectively. In samples 10 and 12, as in the others, two types of particles were present, approximately 5 nm and 300 nm in size, but in preparation 12, high-molecular-weight aggregates were found, which could also be due to a higher temperature during autoclaving, as in preparation 6, but their content was lower (up to 4%). Preparations 1 and 2 and 8 and 9 were very different from the other groups according to the PCA data. Preparations 1 and 2 were prepared without autoclaving; preparation 2 was heated at 70 °C for 15 h. High-molecular-weight aggregates (up to 2%) were found in both preparations, although the size and morphology of the low-molecular-weight particles did not differ from other samples. Preparation 8, prepared without autoclaving and heating, contained all the same characteristic particle types, but its viscosity was one of the lowest. Preparation 9, which was prepared similarly to preparation 8 but included heating at 70 °C for 15 h, differed significantly in morphology, forming a branched network, despite having the same particle size distribution. It should be noted that the steps of sterilizing filtration, autoclaving at different temperatures, and further heating did not affect the chemical composition of the poloxamer.

Thus, the major components in all samples are particles of approximately 5 and 300 nm in size. At a P188 concentration of 10% in the sample, the most homogeneous samples were those prepared by sterilization filtration and autoclaving at 0.9 and 1.1 at. Heating resulted in the appearance of a small amount of high-molecular-weight aggregates. The only more homogeneous preparation was the sample that was heated for 15 h after autoclaving at a lower temperature. Autoclaving at a higher temperature with such a poloxamer concentration resulted in an increase in the amount of aggregates to 17%. At the same time, reducing the poloxamer concentration to 4% and adding 0.6% NaCl seemed to help prevent the formation of large molecular aggregates even after heating for 15 h. On the other hand, in the absence of heating, the content of larger particles increased in the samples of 4% poloxamer with NaCl, and an increase in temperature during autoclaving led to the formation of large particles.

It should be noted that samples 4 and 6 were prepared according to our developed method for obtaining an emulsion of perfluorocarbon compounds [22]. A higher autoclaving temperature was used for sample 6, but both samples were subjected to a heating step after the autoclaving process. Based on our data, these two samples are indeed similar, but sample 4, prepared at a lower autoclaving temperature, has no large aggregates in its composition, unlike sample 6, and its viscosity is much lower than that of sample 6.

A study of the effect of prolonged storage on the composition of the samples showed that long-term storage of samples 2–5 for 14 months resulted in an increase in the proportion of aggregates in them. It is interesting to note that samples 1 and 8 were the most stable, with only minor changes in viscosity, particle distribution, and morphology. In general, more dilute P188 solutions (4%) in the presence of salt were more stable, and their properties did not change as much as those of concentrated solutions. The only exception to this was sample 11.

In Russia, for the preparation of the PFO emulsion Perftoran [23] with a shelf life of 1 year, a 10% aqueous solution of poloxamer 188 is registered as a medicinal product Proxanol. In accordance with the regulatory documentation, this substance was tested for stability during the shelf life according to the following parameters: quantitative composition, pH, color, toxicity, sterility, etc. The values of all parameters were within acceptable limits. A stable emulsion of Perftoran was obtained from this substance. However, properties such as particle size in the poloxamer samples and their morphology were not analyzed. In addition, as mentioned above, the final stage in the preparation of the PFO emulsion is to adjust the concentration of PFO compounds and poloxamer to the required value using a water–salt composition. It was interesting to see how the properties of the poloxamer would change if the water–salt solution was used in the first stage of its preparation. In this work, the properties of a poloxamer solution containing a water–salt composition based on sodium chloride with a P188 concentration of 4%, as in the finished product Perftoran, were studied for the first time. The study is the first part of a general study of technological processes affecting the stability of PFO emulsions with the P188 emulsifier, which will allow us to develop the most optimal method for obtaining stable emulsions.

The results of this study may be useful in the selection of the optimal method for the thermal sterilization of the substances used for the preparation of the PFO emulsions.

## 5. Conclusions

The aim of this study was to investigate the influence of different parameters used to prepare sterile poloxamer 188 solutions on their physicochemical properties. The findings revealed that the particle size and morphology are dependent on the concentration of poloxamer 188, the autoclaving mode, the heating method, and the presence of salt.

The most homogeneous samples with a poloxamer 188 concentration of 10% were obtained by sterile filtration and autoclaving at 0.9 and 1.1 atmospheres. The application of heat resulted in the formation of a minor proportion of high-molecular-weight aggregates.

A concentration of 4% poloxamer 188 and 0.6% NaCl prevented the formation of large aggregates even after heating for 15 h. It is important to note that sterile filtration, autoclaving at different temperatures, and subsequent heating did not significantly affect the chemical composition of poloxamer 188.

Low-concentration (4%) poloxamer 188 solutions in the presence of 0.6% NaCl were more stable during long-term storage than concentrated solutions (10%), where larger aggregates formed. The most stable solutions were the P188 solutions, which were prepared by sterilizing filtration only.

## Figures and Tables

**Figure 1 polymers-17-00062-f001:**
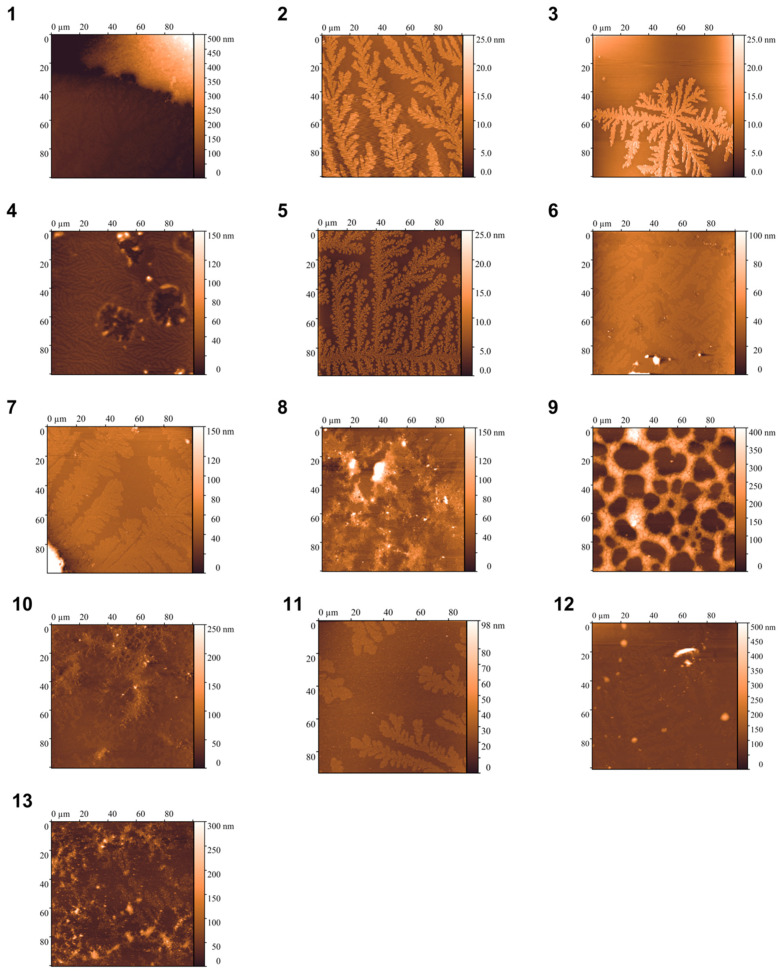
AFM images of one-week samples 1–13 on mica in semi-contact mode (field 100 × 100 µm).

**Figure 2 polymers-17-00062-f002:**
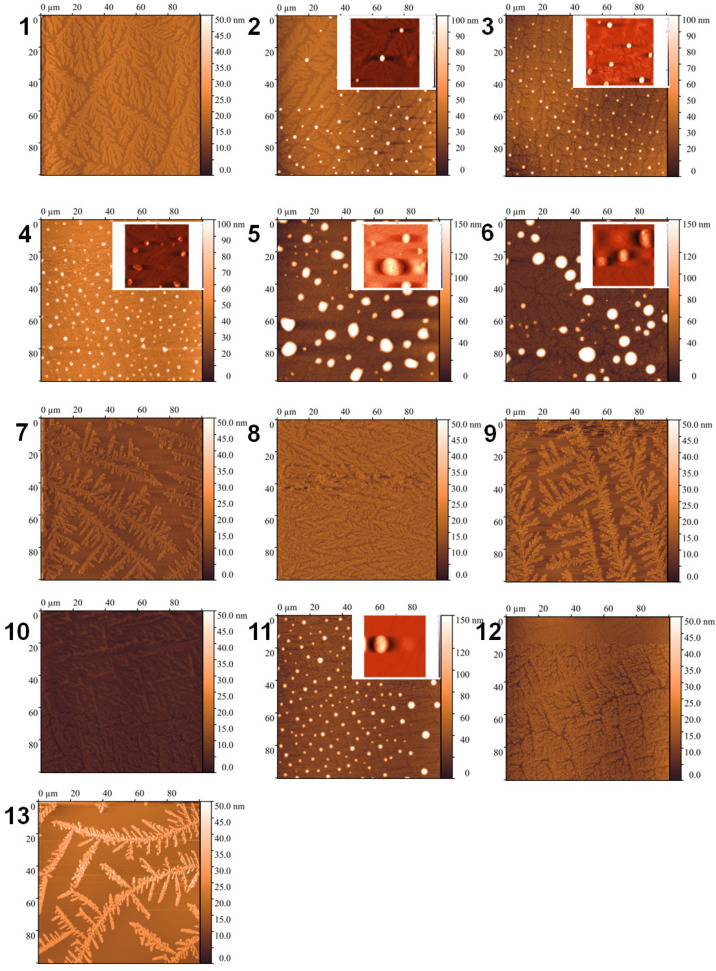
AFM images of samples 1–13 after 14 months on mica in semi-contact mode (field 100 × 100 µm).

**Figure 3 polymers-17-00062-f003:**
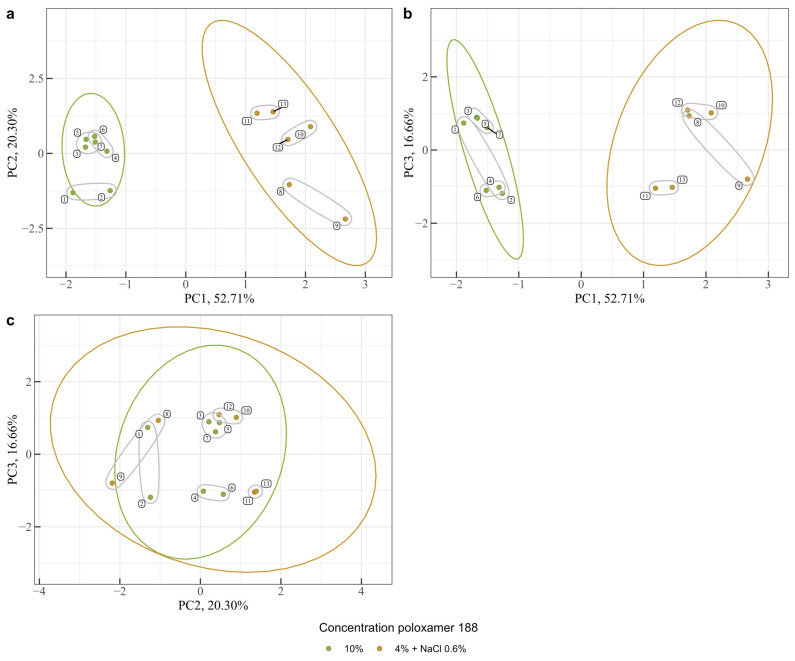
Plots of the first three principal component scores, describing 89.67% of the original variability in the data. (**a**) PC1 vs. PC2, (**b**) PC1 vs. PC3 and (**c**) PC2 vs. PC3

**Figure 4 polymers-17-00062-f004:**
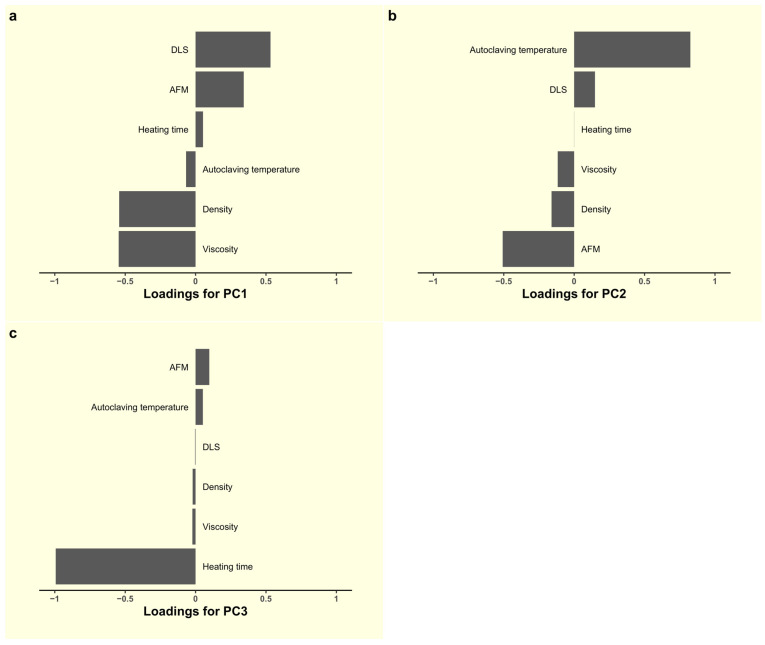
Loadings representing the contribution of each original variable to the corresponding principal component. (**a**) PCA loadings for PC1, (**b**) PCA loadings for PC2 and (**c**) PCA loadings for PC3.

**Figure 5 polymers-17-00062-f005:**
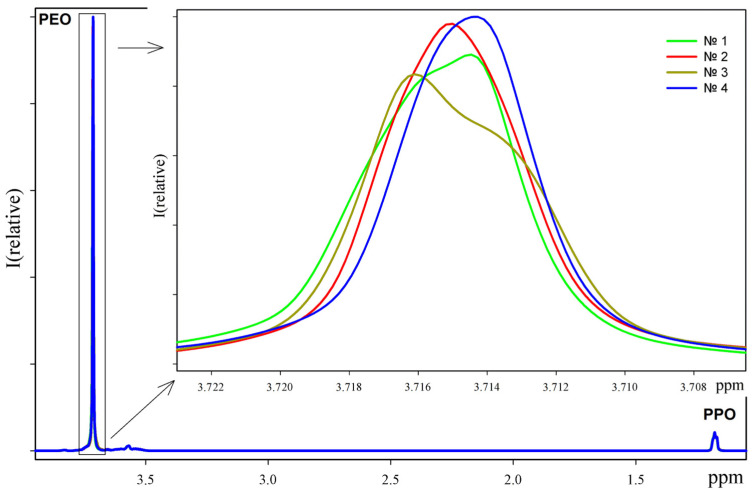
^1^H NMR spectra of poloxamer 1–4 preparations (top right numbers indicate preparation number and spectrum color). The signal at 3.73 ppm is magnified separately on the tab.

**Table 1 polymers-17-00062-t001:** Sample preparation methods.

Sample No.	Concentration of P188, % (w/*w*)	Concentration of NaCl, % (*w*/*w*)	SF ^1^	AM ^2^	Heating time at 70 °C, h
1	10	0	+	-	-
2	10	0	+	-	15
3	10	0	+	0.9 at. (118 °C)	-
4	10	0	+	0.9 at. (118 °C)	15
5	10	0	+	1.1 at. (121 °C)	-
6	10	0	+	1.1 at. (121 °C)	15
7	10	0	+	0.9 at. (118 °C)	2
8	4	0.6	+	-	-
9	4	0.6	+	-	15
10	4	0.6	+	0.9 at. (118 °C)	-
11	4	0.6	+	0.9 at. (118 °C)	15
12	4	0.6	+	1.1 at. (121 °C)	-
13	4	0.6	+	1.1 at. (121 °C)	15

^1^ SF—sterilizing filtration; ^2^ AM—autoclaving mode.

**Table 2 polymers-17-00062-t002:** Viscometric results of the samples studied at 20 °C. The error in the determination of the dynamic viscosity and of the density was not more than 1%.

Sample No.	Density, g/cm^3^	Dynamic Viscosity, mPa 1 Week After Preparation	Dynamic Viscosity, mPa After 14 Months
1	1.014	4.51	4.44
2	1.012	4.38	4.48
3	1.013	4.41	3.72
4	1.013	4.37	3.46
5	1.012	4.45	3.67
6	1.013	4.71	3.82
7	1.012	4.51	4.18
8	1.006	2.27	2.13
9	1.006	2.41	2.35
10	1.006	2.37	2.16
11	1.006	2.33	2.10
12	1.005	2.40	2.04
13	1.005	2.17	1.97

**Table 3 polymers-17-00062-t003:** Particle size distribution of predominant particles in the samples.

Sample No.	1 Week After Preparation		After 14 Months	
	Size, nm	Particle Size Distribution (%)	Size, nm	Particle Size Distribution (%)
1	3.8 ± 0.5	77	3.7 ± 0.1	79.4
	658.1 ± 351.9	19	140.2 ± 29.2	3.1
	4793.0 ± 726.0	2.3	714.5 ± 88.6	17.5
2	3.9 ± 0.6	84.5	3.8 ± 0.1	49.8
	380.4 ± 129.0	14.2	226.1 ± 43.2	10.4
	4772.0 ± 730.4	1.2	777.7 ± 163.5	34.4
			5426.0 ± 402.4	6.8
3	3.8 ± 0.5	85.6	3.7 ± 0.1	57.5
	446.2 ± 126.3	14.4	301.4 ± 87.9	18.3
			1098.9 ± 450.2	20.7
4	3.9 ± 0.7 429.8 ± 134.6	88.1 11.9	3.8 ± 0.2	62.8
			144.9 ± 35.8	11.8
			601.4 ± 237.9	26.0
5	3.8 ± 0.5 380.5 ± 94.2	84.0 16.0	3.7 ± 0.2	57.3
			220.5 ± 126.8	10.3
			794.8 ± 218.6	28.2
6	4.1 ± 0.7	74.9	3.9 ± 0.2	84.5
	247.1 ± 66.0	7.5	356.7 ± 69.8	5.7
	1090.0 ± 643.0	16.2	2995.5 ± 1075.7	9.7
7	3.9 ± 0.6	77.2	3.8 ± 0.2	65.1
	437.4 ± 169.8	19.2	455.5 ± 150.9	15.9
	3818.0 ± 1127.0	3.6	2672.3 ± 991.1	14.7
8	4.6 ± 0.8	89.8	4.5 ± 0.1	88.8
	449.9 ± 128.7	10.2	470.2 ± 102.0	8.1
			4034.0 ± 419.9	3.1
9	4.7 ± 1.0	86.7	4.4 ± 0.1	75.7
	416.9 ± 138.3	13.3	309.5 ± 27.9	21.7
			857.4 ± 180.0	5.1
10	5.1 ± 0.5	75.6	4.3 ± 0.1	73.2
	373.5 ± 134.3	24.4	116.4 ± 41.9	4.9
			600.1 ± 104.9	20.3
11	4.6 ± 1.0	89.0	4.9 ± 0.1	69.0
	451.3 ± 202.6	11.0	192.0 ± 53.5	7.4
			1054.0 ± 383.8	21.2
12	4.5 ± 0.6	86.1	4.2 ± 0.1	78.8
	270.1 ± 81.2	10.3	560.5 ± 233.7	14.2
	3355.0 ± 1207.0	3.6	3895.0 ± 1057.0	4.3
13	4.6 ± 1.0	88.2	4.6 ± 0.1	76.2
	294.9 ± 87.2	11.8	362.5 ± 155.4	13.1
			2680.0 ± 959.9	10.2

## Data Availability

The original contributions presented in this study are included in the article/Appendix A. Further inquiries can be directed to the corresponding author.

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
