# Peer review of "Influence of Parameters Used to Prepare Sterile Solutions of Poloxamer 188 on Their Physicochemical Properties"

_polymers, 2024, doi:10.3390/polym17010062_

Round 1

Reviewer 1 Report

Comments and Suggestions for Authors

The authors work seems helpful to explore various physicochemical properties of Poloxamer 188, but based on the authors existing data, it is not enough to publish as a research paper. It is suggested to improve the paper with following contents:

1 In the introduction section, the authors should explain what role this study has in expanding the use of Poloxamer 188 or exploring its potential applications;

2. Except for the representative AFM images, the authors did not provide any optical photos, SEM images or other information, and the reviewer thought that such reports were incomplete;

3 In addition, reviewer remain concerned about the workload, despite the large number of controls, see no discussion of repeatability, especially long-term stability. For example, based on past experience, triblocopolymer itself has a lot of instability, and it needs to do a lot of repeated experiments when preparing micelles. 

4 It is suggested that the authors discuss the influence of other variable parameters besides two autoclaving modes, especially the influence of high temperature and viscosity on Poloxamer 188.

5 Too few references.

Author Response

Comments 1: In the introduction section, the authors should explain what role this study has in expanding the use of Poloxamer 188 or exploring its potential applications

Response 1: Thank you very much for your comment. Initially, our task was to study the effect of the manufacturing process of the poloxamer preparation for the production of perfluorocarbon blood substitutes based on it. However, as poloxamer P188 has recently been considered as an active substance due to its ability to protect neuronal and non-neuronal cells, to improve cellular functions after exposure to various stress factors, to restore erythrocyte functions, to act as a mitigator after irradiation, with low toxicity and minimal side effects, and is therefore of interest not only in the composition of emulsions, we have included a small fragment about its properties in the introduction.

Comments 2: Except for the representative AFM images, the authors did not provide any optical photos, SEM images or other information, and the reviewer thought that such reports were incomplete;

Response 2: Thank you for your comment. The method available to us was AFM, so we took AFM images with different field sizes. If possible, we would also like to examine our samples by SEM.

Comments 3:  In addition, reviewer remain concerned about the workload, despite the large number of controls, see no discussion of repeatability, especially long-term stability. For example, based on past experience, triblocopolymer itself has a lot of instability, and it needs to do a lot of repeated experiments when preparing micelles

Response 3: Thank you for your comment. As the production of a solution of poloxamer P188 is part of the manufacturing process of blood substitutes, such studies were performed regularly for samples 1-7 and the results were reproducible. Samples 8-13 were tested for the first time. We decided to include in the work a study of changes in solutions of poloxamer P188 after 14 months of storage. It should be noted that the solution of poloxamer Proxanol is registered in Russia as a drug with a shelf life of 1 year, and studies of its stability during the year were conducted (quantitative composition, pH, colour, toxicity, sterility), but a study of changes in the distribution of particles in the solution and a study of their morphology were not conducted.

Comments 4:  It is suggested that the authors discuss the influence of other variable parameters besides two autoclaving modes, especially the influence of high temperature and viscosity on Poloxamer 188.

Response 4: Autoclaving is part of our production process, so we focused on the parameters we were interested in: autoclaving modes and post-autoclaving heating.

Comments 5:  Too few references

Response 5: We tried to improve English as much as possible. Thanks for your comments, we have added references.

Reviewer 2 Report

Comments and Suggestions for Authors

The manuscript is helpful for readers in this area of homogenization of emulsions and consequently merits publication. However, I do have some comments prior to publication:

1. In the abstract, clarify what is the triblock copolymer of poloxamer 188? Heating of the 10% poloxamer samples….temperature? Instruments used? physicochemical properties tested?.

2. In the introduction, please provide a more detailed explanations on the synthesis of Poloxamer 188. Also, some relevant recent works from the literature should be added.

3. Line 52, PFO, provide its full name. Associate the abbreviations with their full names at the first mention, for example line 61 dynamic light scattering (DLS), …etc.

4. Symbol of Litre should be L

5. It would be better providing the figures of the size distribution by DLS.

6. Fig. 4 must be improved.

7. Line 326, correct spectrum to spectra

8. The section of discussion suffers from lack of the relevant explanations, as well as the supporting references.

9. Conclusion is well written.

10. References must be updated, and more relevant references (2022-2024) must be added.

Author Response

Comments 1 In the abstract, clarify what is the triblock copolymer of poloxamer 188? Heating of the 10% poloxamer samples….temperature? Instruments used? physicochemical properties tested?.

Response 1: Thank you for your constructive remarks. We have corrected the abstract according to your instructions.

Comments 2: In the introduction, please provide a more detailed explanations on the synthesis of Poloxamer 188. Also, some relevant recent works from the literature should be added.

Response 2: Thank you for your comment. In our manufacturing process, we do not synthesize poloxamer, but use the commercially available poloxamer 188 (Kolliphor P188 (GND10221B series, BASF, Germany)), the solution preparation of which is a step in the production of blood substitute emulsions. The preparation of the P188 solution was of great interest to us because of the influence of the preparation of the solution on the properties of the P188 solution. Аs had previously been shown the post-autoclave heating step significantly improved the stability of the blood substitutes emulsions.

Comments 3:  Line 52, PFO, provide its full name. Associate the abbreviations with their full names at the first mention, for example line 61 dynamic light scattering (DLS), …etc.

Response 3: We have tried to correct these errors everywhere, thank you for your comments.

Comments 4:  Symbol of Litre should be L

Response 4: Thank you very much. We have corrected this error.

Comments 5:  It would be better providing the figures of the size distribution by DLS.

Response 5: Due to the large number of samples and the fact that the article has been revised to include an analysis of the stability of the samples after 14 months, the DLS data will be presented as a table, but the DLS images will be included as a supplement.

Comments 6:  Fig. 4 must be improved.

Response 6: We have improved Fig.4

Comments 7:  Line 326, correct spectrum to spectra

Response 7: We have corrected this.

Comments 8:  The section of discussion suffers from lack of the relevant explanations, as well as the supporting references.

Response 8: Thank you for your comment. We have rewritten the discussion a little, taking into account the comments.

Comments 9:  Conclusion is well written.

Response 9: Thank you for your kind words about our work.

Comments 10:  References must be updated, and more relevant references (2022-2024) must be added.

Response 10: Thanks for the comment, we have added the references.

Round 2

Reviewer 1 Report

Comments and Suggestions for Authors

Please mark the revised text so that reviewer can trace the revision.

Author Response

Comments 1: Please mark the revised text so that reviewer can trace the revision.

Response 1: Thank you for your constructive remark. We have marked the revised text.

Reviewer 2 Report

Comments and Suggestions for Authors

The authors addressed all comments very well and the quality of the manuscript has been significantly improved. From my side, the current version is suitable for publication by Polymers/MDPI

Author Response

Comments 1: The authors addressed all comments very well and the quality of the manuscript has been significantly improved. From my side, the current version is suitable for publication by Polymers/MDPI.

Response 1: Thank you for your constructive comments that enabled us to improve our manuscript.

Round 3

Reviewer 1 Report

Comments and Suggestions for Authors

I think the paper could be published, and the supporting figures (and related discussions) are suggested to be put into a word file.